# Lack of PKCθ Promotes Regenerative Ability of Muscle Stem Cells in Chronic Muscle Injury

**DOI:** 10.3390/ijms21030932

**Published:** 2020-01-31

**Authors:** Piera Filomena Fiore, Anna Benedetti, Martina Sandonà, Luca Madaro, Marco De Bardi, Valentina Saccone, Pier Lorenzo Puri, Cesare Gargioli, Biliana Lozanoska-Ochser, Marina Bouché

**Affiliations:** 1Department of AHFMO, University of Rome “la Sapienza”, Via A. Scarpa 14, 00161 Rome, Italy; pierafilomena.fiore@opbg.net (P.F.F.); anna.benedetti@uniroma1.it (A.B.); martinasandona2@gmail.com (M.S.); luca.madaro@uniroma1.it (L.M.); biliana.lozanoska-ochser@uniroma1.it (B.L.-O.); 2IRCCS Fondazione Santa Lucia (FSL), e00143 Rome, Italy; m.debardi@hsantalucia.it (M.D.B.); v.saccone@hsantalucia.it (V.S.); 3Department of Life Sciences and Public Health, Università Cattolica del Sacro Cuore, 00168 Rome, Italy; 4Development, Aging and Regeneration Program, Sanford Burnham Prebys Medical Discovery Institute, La Jolla, CA 92037, USA; lpuri@sbpdiscovery.org; 5Department of Biology, Tor Vergata University, 00133 Rome, Italy; Cesare.Gargioli@uniroma2.it

**Keywords:** Duchenne Muscular Dystrophy, *mdx*, muscle satellite cells, Protein Kinase C θ

## Abstract

Duchenne muscular dystrophy (DMD) is a genetic disease characterized by muscle wasting and chronic inflammation, leading to impaired satellite cells (SCs) function and exhaustion of their regenerative capacity. We previously showed that lack of PKCθ in *mdx* mice, a mouse model of DMD, reduces muscle wasting and inflammation, and improves muscle regeneration and performance at early stages of the disease. In this study, we show that muscle regeneration is boosted, and fibrosis reduced in *mdx*θ^−/−^ mice, even at advanced stages of the disease. This phenotype was associated with a higher number of Pax7 positive cells in *mdx*θ^−/−^ muscle compared with *mdx* muscle, during the progression of the disease. Moreover, the expression level of Pax7 and Notch1, the pivotal regulators of SCs self-renewal, were upregulated in SCs isolated from *mdx*θ^−/−^ muscle compared with *mdx* derived SCs. Likewise, the expression of the Notch ligands Delta1 and Jagged1 was higher in *mdx*θ^−/−^ muscle compared with *mdx*. The expression level of Delta1 and Jagged1 was also higher in PKCθ^−/−^ muscle compared with WT muscle following acute injury. In addition, lack of PKCθ prolonged the survival and sustained the differentiation of transplanted myogenic progenitors. Overall, our results suggest that lack of PKCθ promotes muscle repair in dystrophic mice, supporting stem cells survival and maintenance through increased Delta-Notch signaling.

## 1. Introduction

Duchenne muscular dystrophy (DMD) is a severe form of muscular dystrophy caused by lack of dystrophin, leading to membrane instability and increased probability of damage during contraction. As a result, dystrophic muscle is characterized by persistent inflammation, triggered by cycles of degeneration and regeneration. Chronic inflammation, in turn, impairs muscle regeneration and promotes the deposition of fibrotic and adipose tissue, which progressively replace the muscle tissue [1,2,3].

In DMD patients, the necrotic fibers are initially replaced by new fibers generated by satellite cells (SCs), the main myogenic stem cells in adult muscle. Upon muscle injury, SCs become activated, proliferate, and differentiate into myofibers while a minor subset self-renews to replenish the SCs pool [4]. Satellite cell self-renewal preserves the muscle regenerative capability following multiple injuries. Once activated, SCs can generate functionally different daughter cells by asymmetric division: One daughter will undergo differentiation after a variable number of symmetric cell divisions, while the other will return to the quiescent state. The “self-renewing” satellite cell is characterized by stable expression of Paired box 7 (Pax7), while the differentiating one upregulates Myogenic Differentiation1 (MyoD) expression and downregulates Pax7 [5,6].

In dystrophic muscle, the regeneration ability of SCs rapidly declines with age. In *mdx* mice, a DMD animal model, muscle repair after injury is also impaired compared with WT mice, due to chronic inflammation and the exhaustion of the SCs pool [7,8,9,10]. Moreover, the decreased number of satellite cells in dystrophic muscle during aging has been linked to impaired Notch signaling. Notch signaling is involved in regulation of satellite cell activation and self-renewal. Notch 1, 2, and 3 are expressed in quiescent SCs, while muscle fibers are the major source of Notch ligands, such as Delta and Jagged [11,12,13]. Notch activation prevents myogenic differentiation and promotes satellite cell self-renewal, by upregulating Pax7 and inhibiting MyoD [14,15]. In the absence of Notch, SCs undergo accelerated terminal differentiation without self-renewal, resulting in muscle stem cell pool depletion [16].

In *mdx* mice, activation of the Notch pathway rescued the self-renewal ability of satellite cells [17]. Intriguingly, in a canine model of DMD, two Golden Retriever muscular dystrophy (GRMD) dogs, were found to suffer a milder dystrophic phenotype. This milder phenotype was correlated to increased Jagged1 expression, suggesting that promoting Notch signaling may represent a therapeutic approach for DMD in a dystrophin-independent manner [18].

We previously showed that lack or pharmacological inhibition of PKCθ reduced muscle loss and inflammation, and increased muscle regeneration and performance in *mdx* mice. The observed phenotype was primarily due to lack of PKCθ in hematopoietic cells [19,20], and in particular inhibition of early T cells infiltration in dystrophic muscle [21]. However, PKCθ is also expressed in muscle, where it modulates several signalling pathways involved in foetal and early post-natal tissue growth and maturation [22,23,24]. Intriguingly, we observed enhanced muscle regeneration in dystrophic muscle lacking PKCθ, compared to *mdx*, and reduced muscle necrosis and fibrosis, suggesting that PKCθ might also regulate SC function during disease progression. Here we show that lack of PKCθ in *mdx* mice improves the survival and maintenance of both endogenous and transplanted stem cells, most likely by promoting Notch signaling.

## 2. Results

### 2.1. Lack of PKC*θ* in Mdx Mice Boosts Muscle Regeneration While Reducing Muscle Fiber Degeneration

The progression of the dystrophic pathology in mdx mice follows distinct phases of muscle degeneration and regeneration. Up until two weeks of age, the muscle morphology is similar in *mdx* and WT mice. Muscle fiber degeneration in mdx mice becomes evident at around 3 weeks of age and peaks at 4 weeks. The high level of muscle degeneration is then followed by high level of muscle regeneration. By 3 months of age, the cycles of both degeneration and regeneration are attenuated, and the skeletal muscle enters a stable phase [25,26].

To further understand the eventual role of PKCθ in regulating muscle regeneration and satellite cells function in dystrophic muscle, we first analyzed the histo-pathological features, in terms of the extent of muscle degeneration and regeneration, in *mdx*θ^−/−^ mice compared to *mdx*, during the progression of the disease, from 1 to 12 months of age. Muscle degeneration was evaluated in tibialis anterior (TA) muscle by H&E (Appendix A), while muscle regeneration was evaluated by embryonic myosin heavy chain (eMHC) immunostaining (Appendix A). The results, summarized in Figure 1A, show a peak in muscle damage at 4 weeks as expected, with a sharp drop by 6 weeks of age; after this age, muscle damage continued to slowly decrease, up to 12 months of age. Similarly, the high level of muscle regeneration (Figure 1B) observed at 4 weeks of age, dropped drastically by 6 weeks and remained low until 12 months of age. By contrast, the level of muscle damage was much lower in 4-week-old *mdx* mice lacking PKCθ compared to *mdx*, as previously shown [22], and this low level was maintained throughout the ages studied, besides a slight increase at 6 months of age. Consistently, the level of muscle regeneration was lower in 4-week-old *mdx*/θ^−/−^ compared to *mdx*, but this level was maintained throughout the 12-month time period. Expressing these results as the ratio of regenerating area over damaged area (Figure 1C), we found that in *mdx*θ^−/−^ muscle the ratio is higher at all the ages examined, suggesting that the regeneration process is boosted compared to *mdx* during the progression of the disease, independently from the level of muscle damage.

During disease progression, chronic damage and inflammation are known to prevent adequate regeneration leading to increased ECM deposition and fibrotic tissue accumulation, which is one of the most deleterious aspects of DMD. The Masson’s trichrome staining of TA sections (Figure 1) showed that the increased collagen deposition observed in mdx mice during the progression of the disease, compared to WT mice, is significantly reduced when PKCθ is absent, at all the ages examined. These findings suggest that lack of PKCθ reduces muscle necrosis and fibrosis and improves regeneration.

### 2.2. Dystrophic Muscle Repair After Injury is Enhanced in The Absence of PKCθ

The repeated cycles of degeneration and regeneration and the hostile dystrophic environment are believed to exhaust the regenerative capacity of SCs. Indeed, after acute injury, the muscle repair is impaired in *mdx* mice compared with WT mice [27] and worsens over time [7]. Therefore, we wondered whether lack of PKCθ may improve the regenerative ability of dystrophic muscle following injury. TA muscle of 6-month-old *mdx* and *mdx*θ^−/−^ mice were injured by intra-muscular cardiotoxin (CTX) injection. Age- and sex-matched WT mice were used for comparison. The mice were sacrificed 7 days after CTX injection, and the number of regenerating myofibers, identified as centrally nucleated fibers, was counted in H&E stained sections of TA muscles (Figure 2A). As shown in Figure 2B, a significantly higher number of regenerating fibers was found in TA muscle derived from *mdx*θ^−/−^ mice compared with *mdx*, almost restoring the number of regenerating fibers found in injured WT muscle. Skeletal muscle repair following injury requires the deposition, remodeling, and reorganization of ECM. ECM deposition occurs within a week post-injury, and it can continue for several weeks to ensure complete repair. However, in the case of chronic injuries, such as in dystrophic muscle, the newly generated ECM can turn into scar tissue [28]. As shown in Figure 2C, 7 days after CTX injury, the amount of ECM deposition in the injured area was greater in *mdx* mice compared to WT mice, as expected. Interestingly, lack of PKCθ significantly reduced ECM deposition in mdx mice. Together, these results suggest that in the absence of PKCθ, dystrophic muscle preserves the ability of skeletal muscle to repair the damaged area.

### 2.3. Lack of PKCθ in Mdx Mice Preserves the Self-Renewal Ability of Satellite Cells

Given the reduction of muscle damage during the early stages of the disease in *mdx*θ^−/−^, we wondered whether the enhanced repair ability was due to a more efficient maintenance of the SC pool compared with *mdx* mice. Indeed, in muscular dystrophy the continuous rounds of damage can exhaust the self-renewal ability of satellite cells [27]. In the early stages of the muscle repair process, quiescent SCs migrate to the site of injury and proliferate. Following the proliferation phase, a subset of SCs undergoes myogenic differentiation, downregulating Pax7 expression and upregulating MyoD expression. Another subset of proliferating SCs maintains Pax7 expression, and returns to a quiescent state, ensuring renewal of the SC pool [29,30,31]. First, TA section from *mdx* and *mdx*θ^−/−^ mice were immuno-stained for Pax7 and Laminin expression at different ages (Appendix A). The counting of the number of total Pax7 positive cells in each condition revealed that it was significantly higher in TA muscle from *mdx*θ^−/−^ compared to *mdx* mice at all the ages examined (Figure 3A). Next, we performed qRT-PCR analysis and found that the level of expression of Myogenin at later stages of the disease (12 months of age), was higher in *mdx*θ^−/−^ mice compared to *mdx* mice (Figure 3B), suggesting that lack of PKCθ improves SCs ability to repair damaged muscle. Myogenin expression is upregulated in differentiating satellite cells, and it is considered a marker of muscle regeneration [32]. This increase was not observed in muscle derived from PKCθ^−/−^ as compared with WT, suggesting that it is a consequence of lack of PKCθ specifically in a dystrophic environment (Figure 3B).

Together, these results suggest that lack of PKCθ preserves the SCs pool in dystrophic muscle even at advanced ages, without affecting the balance between self-renewal and myogenic commitment.

### 2.4. The Self-Renewal Ability of Satellite Cells in *mdx*θ^−/−^ Muscle is Maintained and Supported by Up-Regulation of Notch-Signaling

To further characterize the effect of lack of PKCθ on dystrophic SCs activity, FACS-isolated SCs from limb muscle of 6-week-old *mdx* and *mdx*θ^−/−^ mice were cultured in vitro. We chose 6 weeks of age because this is the age when muscle regeneration starts to decline in *mdx* muscle. After 62 h in culture, the cells were analyzed for the expression of Pax7 and MyoD by immunofluorescence. Based on the expression of Pax7 and MyoD, cultured satellite cells are defined as self-renewing (Pax7^+^/MyoD^−^), activated (Pax7^+^/MyoD^+^) or differentiating (Pax7^−^/MyoD^+^) [5,33]. As shown in Figure 4, the percentage of Pax7^+^/MyoD^−^ “self-renewing” cells over the total number of Pax7^+^cells, was higher in SCs derived from *mdx*θ^−/−^ muscle compared to *mdx*.

Next, we analyzed the expression level of genes involved in SC self-renewal (Pax7, MyoD, Notch1-2-3, Delta1, and Numb) in freshly FACS-sorted SCs, by qRT-PCR analysis. As shown in Figure 5A, Pax7 and Notch1 level of expression was significantly higher in SCs freshly isolated from *mdx*θ^−/−^ mice compared with those isolated from *mdx* mice. Notch signaling in SCs is activated by the binding of the Notch receptor to the Jagged1 and Delta1 ligands expressed on the myofiber surface. It is known that Delta1 expression decreases in aged and dystrophic muscle [7,34]. As shown in Figure 5B, Delta1 level of expression is significantly higher in TA muscle derived from *mdx*θ^−/−^ compared to *mdx* muscle at 6 and 12 weeks, as well as at 12 months of age (Figure 5B). Also, Jagged1 level of expression is higher in TA muscle derived from 6 week and 12 months old *mdx*θ^−/−^ mice, compared to muscle from age-matched *mdx* mice (Figure 5B). The high level of expression of Notch ligands in *mdx*θ^−/−^ muscle may suggest an active role of myofibers to boost the self-renewal process.

Interestingly, we found that, although Delta1 and Jagged1 level of expression was similar in healthy WT and PKCθ^−/−^ non-dystrophic muscle (Appendix A), upon acute injury, at day 3 after CTX-injection, the level of expression of Jagged1 was significantly higher in PKCθ^−/−^ muscle compared to WT. Conversely, the expression level of Delta1 was higher in PKCθ^−/−^ muscle compared to WT at day 7 after injury (Figure 5C). These results suggest that the increased Notch ligands expression observed in injured muscle lacking PKCθ might favor SCs self-renewal, even in non-dystrophic setting.

### 2.5. Lack of PKCθ Supports Transplanted Stem Cells Survival and Differentiation in Injured and Dystrophic Muscle

To further verify whether lack of PKCθ in muscle imparts a regenerative advantage, we analyzed the survival and engraftment of transplanted stem cells in both dystrophic and non-dystrophic mice lacking PKCθ. To this aim, we used mouse Mesoangioblasts (MABs), myogenic progenitor cells, as exogenous stem cells. Indeed, Notch signaling has been shown to potentiate mesoangioblasts (MABs)-driven regeneration in vivo [35].

Mouse nLacZ-expressing MABs (5 × 10^5^) [36] were transplanted via intra-muscular injection into TA muscle of WT and PKCθ^−/−^mice, injured with CTX 24 h previously. The mice were sacrificed 3, 7, and 14 days after transplantation. As shown in Figure 6A, the number of nLacZ-MABs within the injured muscle was similar in WT and PKCθ^−/−^ mice up to 7 days after transplantation. Interestingly, 14 days after transplantation, nLacZ-MABs were detectable in TA muscle derived from PKCθ^−/−^ mice, but not in WT. Moreover, although resolution of regeneration was slower in PKCθ^−/−^ mice, as already described [22], LacZ positive nuclei were localized within muscle fibers, showing that the transplanted MABs contributed to the formation of regenerating myofibers.

We then analyzed the survival of transplanted MABs in dystrophic muscle. MABs were transplanted via intra-muscular injection in TA muscle of 6-week-old *mdx* and *mdx*θ^−/−^ mice. The mice were sacrificed 3, 7, and 14 days after transplantation. As in the non-dystrophic background, the number of nLacZ/MABs within the muscle was similar in *mdx* and *mdx*θ^−/−^ mice 3 and 7 days after transplantation. Interestingly, while LacZ positive nuclei in *mdx* muscle were mostly localized in the interstitium, in *mdx*θ^−/−^ muscle they were mostly found within myofibers, even after only 7 days after transplantation (Figure 6B). These results suggest that the muscle environment in *mdx*θ^−/−^ mice improves MABs ability to fuse with muscle fibers compared to *mdx* mice. Importantly, 14 days after transplantation, nLacZ/MABs were detected in muscle derived from *mdx*θ^−/−^ mice, but not from *mdx* mice. Together, these results demonstrate that lack of PKCθ prolongs the survival and supports differentiation and engraftment of transplanted MABs.

## 3. Discussion

The present study shows that lack of PKCθ in *mdx* mice improves muscle repair and promotes satellite cell pool maintenance. We previously showed that lack of PKCθ reduced muscle damage and inflammation and increased muscle performance in 8-week-old *mdx* mice [19]. Here, we show that the high level of muscle damage observed in *mdx* mice is almost blunted in *mdx*θ^−/−^ mice throughout the late disease stages. Moreover, collagen deposition was reduced in *mdx*θ^−/−^ compared to *mdx* mice. Surprisingly, the area of regenerating myofibers was similar between the two genotypes, at all ages examined, suggesting that lack of PKCθ promotes muscle regeneration in dystrophic muscle. Increased regeneration was also observed after acute injury, supporting the hypothesis that the myogenic ability of endogenous cell populations is maintained and promoted in the absence of PKCθ. These results might appear in contrast to our previous observation that lack of PKCθ delayed muscle repair after freeze-injury in a non-dystrophic background. However, the observed delay in muscle repair was not associated with a reduced differentiation activity of satellite cells, nor with changes in cell growth rate, but with a delay in the addition of fusing cells to regenerating myofibers, a process known as “secondary fusion” [22]. On the other hand, in the dystrophic background, compromised muscle repair is due to impaired satellite cells activation and maintenance. Thus, the different muscle regenerative abilities of *mdx* and *mdx*θ^−/−^ could be due to the different behavior of satellite cells due to the different quality of the environment. We show that lack of PKCθ prevents the early peak of muscle degeneration in *mdx*, which, in turn, may prevent the exhaustion of satellite cell pool and preserve their self-renewal ability. Indeed, the number of Pax7 positive cells in muscle is higher in *mdx*θ^−/−^ mice compared to *mdx* mice, and the level of Pax7 and Notch1 expression in satellite cells isolated from *mdx*θ^−/−^ muscle is higher than in SCs from *mdx*. Notch1 signaling is required for the maintenance of the regenerative ability and self-renewal of satellite cells. It is well known that Notch signaling declines in aged mice and in muscular dystrophy [7,34]. Notch signaling is activated by the binding of Delta or Jagged ligands to Notch receptors at the level of the cell membrane. After injury, Notch ligands are upregulated on the myofiber surface, suggesting that the activation of Notch signaling is crucial during muscle repair. Satellite cells are directly associated with the myofiber surface, and Notch ligands on myofibers should activate Notch signaling in satellite cells. Interestingly, we found that in *mdx*θ^−/−^ muscle, Delta1 and Jagged1 expression is higher than in *mdx* muscle, suggesting that the myofibers could support Notch signaling in satellite cells. In basal conditions, PKCθ does not affect Delta1 and Jagged1 expression level; in fact, their expression is similar in WT and PKCθ^−/−^ muscle. In contrast, after acute injury, the level of Delta1 expression is higher in PKCθ^−/−^ muscle compared to WT muscle, suggesting that PKCθ could directly affect Delta1 and Jagged1 expression level after injury. This observation suggests that the up regulation of Delta1 and Jagged1 expression is due to lack of PKCθ in myofibers and not to different muscle environment. In fact, PKCθ is crucial for the adaptive immune response in chronic muscle injury where T cells are involved but, in acute muscle injury, lack of PKCθ does not alter the innate immune response (myeloid cell recruitment) that is important during the regenerating process [21].

Interestingly, lack of PKCθ in *mdx* promotes not only the regenerative activity of endogenous myogenic cell populations, but also the regenerative activity of exogenous transplanted stem cells. In fact, lack of PKCθ in *mdx* muscle improved MABs survival and differentiation in myofibers. Even at 14 days after transplantation, MABs were still detectable in *mdx*θ^−/−^ muscle and were incorporated into myofibers; by contrast, in *mdx* muscle MABs were detectable up to seven days post-transplantation, and mostly in the interstitial space. It is conceivable that lack of PKCθ modifies the inflammatory environment, thus favoring MABs survival and differentiation ability. Moreover, since PKCθ is involved in allograft responses, its lack may partly prevent cell rejection. Our mice were not immune-suppressed, and, although the MABs used derived from the same genetic background, they express beta-gal, which is highly immunogenic. Further, the improved MABs myogenic differentiation within *mdx*θ^−/−^ muscle, may also depend, at least in part, to the observed increase in Delta1 expression on myofibers. Indeed, the Dll1-Notch1 axis regulates the myogenic potential of MABs and ameliorates in vivo MAB-driven regeneration [35].

Together, our results suggest that lack of PKCθ in dystrophic muscle creates a more favorable environment for both endogenous and exogenous cell populations to contribute to muscle maintenance and repair. Although this phenotype is partly dependent on modifications in the quality of the immune response, the results shown here suggest that PKCθ activity might also directly regulate muscle cell populations phenotype and behavior, contributing to the exhaustion of the satellite cell active pool. Finally, these findings improve our understanding of the molecular mechanisms underlying satellite cell pool maintenance in muscular dystrophy paving the way for the design of efficient pharmacological therapeutic strategies aimed at enhancing muscle repair.

## 4. Materials and Methods

### 4.1. Animal Models

PKCθ^−/−^ mice (C57BL/6J background) were previously described [22,37]. *Mdx* mice (C57BL/10ScSn-Dmdmdx/J) were purchased from Jackson laboratory (Bar Harbor, Main, USA) and *mdx* θ^−/−^ transgenic mice were generated in our laboratory (C57BL/6j-C57BL/10ScSn background) [19]. C57BL/10ScSn control mice were purchased from Jackson laboratory. Only males were used. The animals were housed in the Histology Department–accredited animal facility. All the procedures were approved by the Italian Ministry for Health and were conducted according to the U.S. National Institutes of Health (NIH) guidelines.

### 4.2. Muscle Injury Procedure

To induce muscle injury, 10 μL of 10 μmol cardiotoxin solution (Sigma-Aldrich, Saint Louis, MI, USA) was injected in two different area of the TA using a 30 Gauge micro syringe.

### 4.3. Mesangioblast Transplantation

Animals were anesthetized with Avertin (Sigma-Aldrich). Intramuscular injection was performed in TA with 5 × 10^5^ cells/50 μL saline solution, using 30-gauge needles.

### 4.4. Histochemistry and Immunofluorescence Analyses

For histochemistry and immunofluorescence analyses, TA muscles were dissected, embedded in tissue-freezing medium (Leica, Richmond, IL, USA), and snap-frozen in liquid nitrogen-cooled isopentane. Frozen muscle was cut into 7-μm cryosections. For histological analysis, the sections were stained with hematoxylin/eosin or with Masson’s trichrome stain (both from Sigma-Aldrich); for X-galactosidase staining, cryosections were fixed in 0.1% glutaraldehyde, β-galactosidase (βGAL) activity in mice transplanted TA muscles was revealed in transverse cryosections using X-galactosidase (Qiagen, Hilden, Germany). For immunostaining, permeabilization in methanol (6 min at 220 °C) was performed on cryosections after fixation. A citric acid antigen retrieval protocol was used to facilitate PAX7 staining [33]. In brief, slides were heated in a 90 °C solution of 0.01Mcitric acid (pH 6) for 10 min and followed by 2% bovine serum albumin block in PBS. Sections were incubated with primary antibodies for Pax7 (DSHB, University of Iowa, Iowa City, USA; 1:20), Laminin (Sigma-Aldrich; 1:100). Antibody binding was revealed using species-specific secondary antibodies coupled to Alexa Fluor 488 (Life Technologies, Carlsbad, CA, USA), Cy3or Cy5 (Jackson Immunoresearch, West Grove, PA, USA). Nuclei were counterstained with Hoechst 33342 (Thermo Fisher Scientific, Waltham, MA, USA).

The sections were photographed in a Zeiss Axioskop 2 Plusfluorescence microscope, and the images were analyzed using Image J software.

### 4.5. Cell Cultures

Muscle satellite cells (SCs) were prepared from limb skeletal muscles, as previously described [38]. Briefly, dissected muscles were chopped and then digested with 2 μg/mL Collagenase A, 2.4 U/mL Dispase I, 10 ng/mL DNase I (all from F. Hoffmann-La Roche Ltd., Basel, Switzerland), 0.4 mM CaCl2, and 5 mM MgCl2 for 90 min at 37°C. After digestion, the obtained cell suspension was washed with Dulbecco’s phosphate-buffered saline (DPBS) containing 0,2% BSA and then filtered through 70 and then 40 μm cell strainer. Cells were incubated with the mix of the appropriate primary antibodies (10 ng/mL) CD31-PacificBlue (Invitrogen, Carisbad, CA, USA), CD45-eFluor450, Ter119-eFluor450, CD34-Biotin (all from eBioscience, Wien, Österreich), CD11b-Pacific blue, Sca-1-FITC (BD Bioscience, San Jose, CA, USA), and α7integrin-APC (kindly provided by Dr. Fabio Rossi) for 30 min on ice. A subsequent incubation, 30 min on ice, with Streptavidin-PE-Cy7 (1/500; BD Bioscience) was performed. Cells were finally washed and resuspended in HBSS containing 0.2% *w*/*v* BSA and 1% *v*/*v* Penicillin–Streptomycin. Cell sorting were performed on a Beckman Coulter MoFlo High Speed Sorter (Software Summit V4.3.01). SCs were isolated as Ter119−/CD45−/CD31−/CD34+/α7-integrin+/Sca-1− cells. We were performed flow cytometry analysis through FlowJo software (Software FlowJo, LLC, Treestar, CA, USA, v. 10.4.2). SCs were grown on gelatin-coated dishes, in GM (DMEM containing 20% horse serum, HS, 3% chick embryo extract, EE, all from Invitrogen, Carlsbad, CA) in a humidified 5% CO_2_ atmosphere at 37 °C.

nLacZMabs: nLacZ-Mabs derived from C57/bl6 mice [36], were grown in Dulbecco’s modified Eagle medium (DMEM) supplemented with 10% FCS (both from Gibco; Invitrogen, Carlsbad, CA, USA) in a humidified 5% CO2 atmosphere at 37 °C.

### 4.6. RNA Extraction, cDNA Synthesis and qRT-PCR

For RNA isolation from cells, the cells were collected in TRIsureTM (Bioline, London, UK) according to the manufacturer’s instructions. For RNA preparation from muscle, the muscles were homogenized in TRIsure with the ULTRA-TURRAX T25 (IKA^®^-Werke GmbH & Co. KG, Staufen, Germany) and syringed 4 times with a 21G needle syringe. The RNA was converted in cDNA by a RT PCR. For the RT PCR, the High-Capacity cDNA RT kit from Applied Biosystem (Foster City, CA, USA) was used. For RT-qPCR analysis, the BiolineSensiMixTM SYBR Low-ROX Kit (Bioline) was used, following the manufacturer’s protocol. For data analysis, the 75000 Software v2.0.5, provided by Applied Biosystem, was used. Primer list are reported in Appendix A.

### 4.7. Statistical Analysis

Sample size used is indicated in the legend of each Figure. Quantitative data are presented as means ± SD or means ± SEM of at least three different experiments. Statistical analysis to determine significance was performed using unpaired Student’s *t* tests or one-way ANOVA with Tukey’s multiple comparisons, when appropriate. Differences were considered statistically significant at the *p* ≤ 0.05 level.

## 5. Conclusions

In conclusion, our results suggest that PKCθ activity directly regulate muscle cell populations phenotype and behavior, contributing to the exhaustion of the satellite cell active pool in pathological conditions, such as in muscular dystrophy. Indeed, its lack in dystrophic muscle creates a more favorable environment for both endogenous and exogenous cell populations, promoting muscle maintenance and repair. These findings open new and exciting perspectives to the design of novel pharmacological strategies aimed at enhancing muscle repair in pathological conditions.

## Figures and Tables

**Figure 1 ijms-21-00932-f001:**
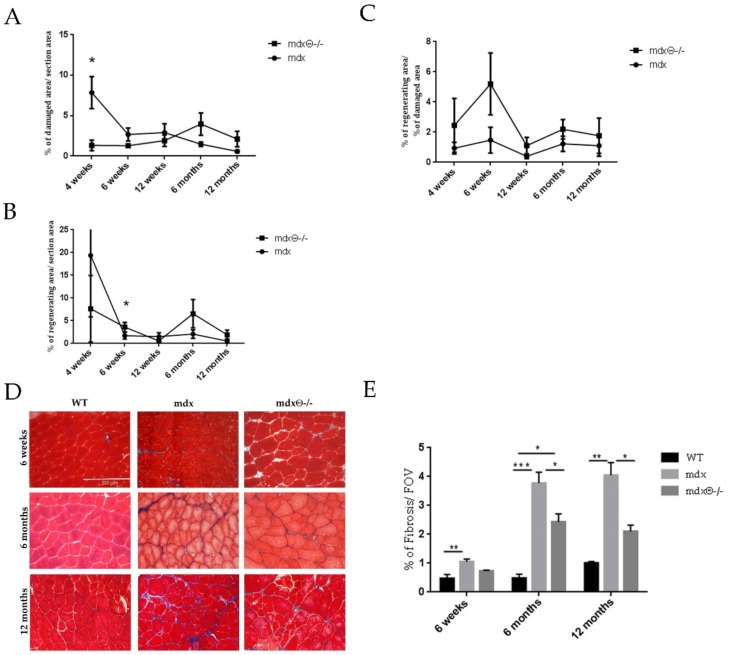
(**A**) Muscle degeneration level evaluated in *mdx* or *mdx/θ^−/−^* mice at the indicated ages, quantified as the percentage of damaged area over the total area in H&E stained TA cryosections. (**B**) Muscle regeneration as in A, quantified as the percentage of eMHC positive area over total area of TA cryosections. (**C**) Ratio of regenerating area over damaged area determined in *mdx* and *mdx*/θ^−/−^ mice at each age evaluated. The results are mean ± SEM (*n* = 4–5/age/genotype); * *p* < 0.05 two-tailed Student’s *t*-test). (**D**) Representative images of Masson’s trichrome staining of cryosections derived from WT/bl10, *mdx* and *mdx*θ^−/−^ TA muscle at 6 weeks, 6 and 12 months (bar = 200 µm). (**E**) Fibrosis was quantified as the percentage of Masson’s trichrome positive area per field of view in TA section. Statistical significance was determined by a one-way ANOVA using a Tukey’s post-test (* *p* < 0.05, ** *p* < 0.01, *** *p* < 0.001 means ± SD).

**Figure 2 ijms-21-00932-f002:**
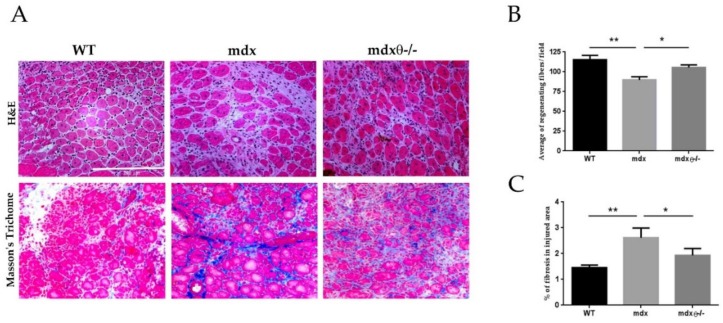
(**A**) Representative image of TA stained with H&E (**upper panels**) and Masson’s trichrome staining (**lower panels**) of 6- month-old WT/bl10 (*n* = 3), *mdx* and *mdx*θ^−/−^ mice (*n* = 5/genotype), as indicated, at day 7 after CTX injury (bar = 200µm). (**B**) Average of number of regenerating fibers and (**C**) Quantification of collagen deposition per field of view. Statistical significance was determined by a one-way ANOVA using a Tukey’s post-test (* *p* < 0.05, ** *p* < 0.01, means ± SD).

**Figure 3 ijms-21-00932-f003:**
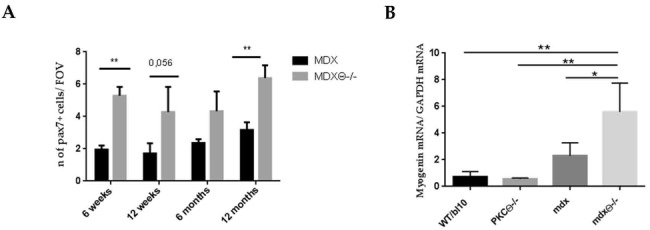
(**A**) Number of Pax7+ cells in TA section of *mdx* and *mdx*θ^−/−^ mice at 6 and 12 weeks, and 6 and 12 months of age. The number of Pax7^+^ cells was calculated by counting the number of PAX7^+^ cells per field of view in immune-stained sections. The results are the means ± SD (*n* = 3/ genotype/ age (* *p* < 0.05, ** *p* < 0.01, two-tailed Student’s *t*-test). (**B**) Myogenin mRNA levels in WT (*n* = 2), PKCθ^−/−^ (*n* = 3), *mdx* (*n* = 5) and *mdx*θ^−/−^ (*n* = 4) TA muscle from 12 months old mice. The values were normalized vs. GAPDH mRNA level of expression. Statistical significance was determined by a one-way ANOVA using a Tukey’s post-test (* *p* < 0.05, means ± SD).

**Figure 4 ijms-21-00932-f004:**
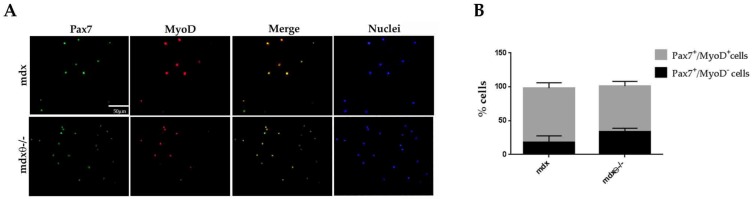
(**A**) Representative image of Pax7 (green) and MyoD (red) immunofluorescence of SCs, isolated from mdx and *mdx*θ^−/−^ muscle and cultured for 62 h. Nuclei were counterstained with Hoechst (blue) (bar = 50 µm). (**B**) Percentages of Pax7^+^/MyoD^−^ and Pax7^+^/MyoD^+^ cells over the total number of nuclei. The results are mean ±S D) (*n* = 3/genotype).

**Figure 5 ijms-21-00932-f005:**
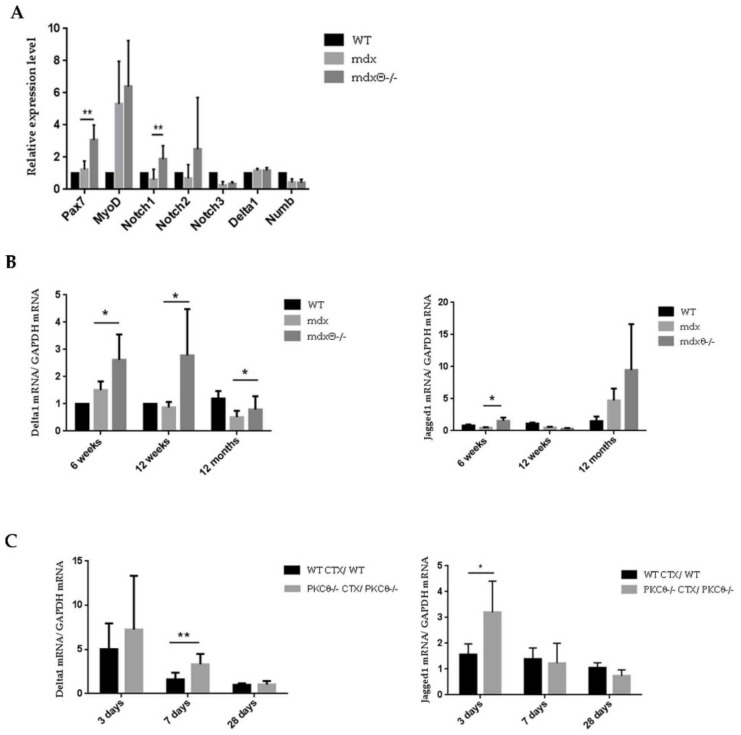
(**A**) Sorted SCs were analyzed for the level of expression of Pax7, MyoD, Notch receptors, Delta1, and Numb by qRT-PCR. RNA expression level in mdx and *mdx*θ^−/−^ SCs was normalized against GAPDH. Samples represent three pooled RNA from SCs sorted from three mice for each sample. (* *p* < 0.05, ** *p* < 0.01, two-tailed Student’s *t*-test). (**B**) Delta1 and Jagged1 mRNA level in mdx and *mdx*θ^−/−^ TA (*n* = 4–7/genotype/age). (* *p* < 0.05, two-tailed Student’s *t*-test). (**C**) Ratio of Delta1 and Jagged1 mRNA level between injured and contralateral uninjured TA muscle in WT/bl6 and PKCθ^−/−^ at 3, 7 and 28 days after CTX injury (*n* = 4–6/ genotype).

**Figure 6 ijms-21-00932-f006:**
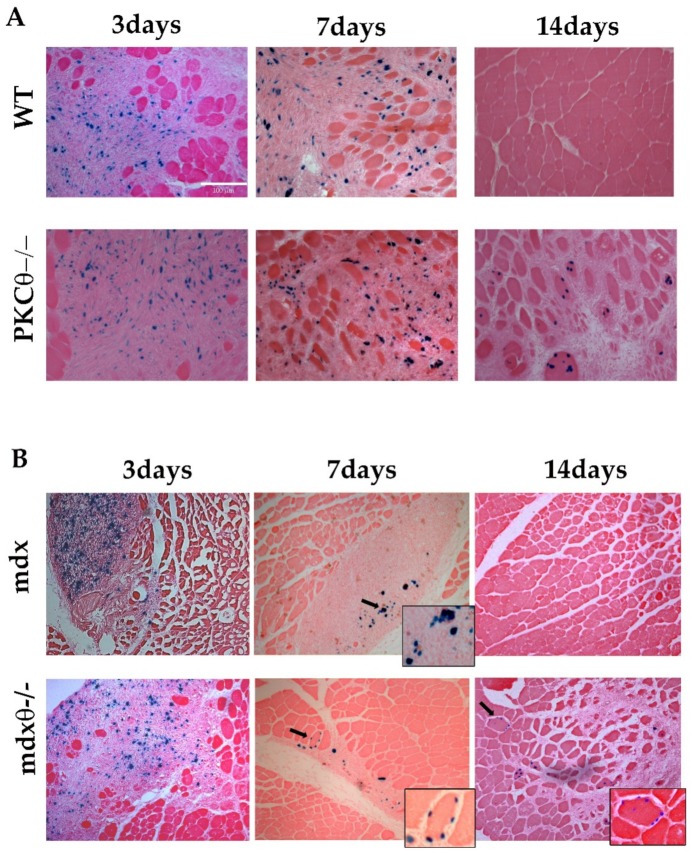
(**A**) Representative cryosections of TA muscle derived from WT and PKCθ^−/−^ sacrificed 3, 7, or 14 days after transplantation (*n* = 3–5/genotype, each time point), as indicated, and processed for X-gal activity (bar = 100µm). (**B**) Representative cryosections of TA muscle derived from *mdx* and *mdx*θ^−/−^ sacrificed 3, 7 or 14 days after transplantation, (*n* = 3–5/genotype, each time point), as indicated, and processed for X-gal activity. Small side pictures show higher magnification of parts of the sections. Arrows indicate X-gal positive nuclei included within some fibers in PKCθ-null mice, in both non-dystrophic and dystrophic background.

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
