# Peer review of "Lack of PKCθ Promotes Regenerative Ability of Muscle Stem Cells in Chronic Muscle Injury"

_ijms, 2020, doi:10.3390/ijms21030932_

Round 1
Reviewer 1 Report
The manuscript by Fiore et al unravels an interesting roleof PKCθ in the regulation of muscle stem cell proliferation as opposed to differentiation. Stem cell exahaustion due to excess proliferation has been suggested as a pathomechanism of muscular dystrophies. Thus, the identification of a potential molecular target of intervention as PKCθ is of great interest for research in Duchenne and other muscular dystrophies. The authors, by taking advantage of the well-known mdx mouse model, show involvement of Notch signaling, which can be modulated by PKCθ, in the pathomechanism of Duchenne muscular dystrophy.
Data are well presented.
I suggest some additional experiment or parameter demonstrating shift of myoblasts or other muscle cell precursors towards proliferation in the absence of PKCθ activity. For instance, differentiation of human myoblasts after PKCθ inhibition could be interesting.
Some infos on on mouse muscle motor function in mice lacking PKCθ could be added.
Author Response
The manuscript by Fiore et al unravels an interesting role of PKCθ in the regulation of muscle stem cell proliferation as opposed to differentiation. Stem cell exahaustion due to excess proliferation has been suggested as a pathomechanism of muscular dystrophies. Thus, the identification of a potential molecular target of intervention as PKCθ is of great interest for research in Duchenne and other muscular dystrophies. The authors, by taking advantage of the well-known mdx mouse model, show involvement of Notch signaling, which can be modulated by PKCθ, in the pathomechanism of Duchenne muscular dystrophy.
Data are well presented.
We thank the Reviewer for the positive comment
I suggest some additional experiment or parameter demonstrating shift of myoblasts or other muscle cell precursors towards proliferation in the absence of PKCθ activity.
We agree with the Reviewer that this is an interesting issue. Indeed, as we now mention in the text (page 8, line 266), we have already showed that lack of PKCθ does not change growth rate in myoblasts (Madaro et al. 2010). This result was also recently confirmed by a CFSE cytofluorimetric analysis (unplublished observation).
For instance, differentiation of human myoblasts after PKCθ inhibition could be interesting.
We agree with the Reviewer that it would be very interesting to analyse the effect of PKCθ inhibition on human myoblasts. Unfortunately, those cells are not easily available; thus, we hope to be able to investigate this aspect in a near future.
Some infos on mouse muscle function in mice lacking PKCθ could be added.
We agree that this is an interesting issue; we addressed this aspect in our previous studies (Madaro et al, 2012; Marrocco et al. 2017), as it is mentioned in the Introduction (page 2, line 68).
Reviewer 2 Report
Dear authors, i have a few comments on your paper. I think it is in general terms well written and presented. One of the major limitations of this work is the replicates number . I believe this limitation must be indicated in the discussion. In addition, it would be appropriate to add the sample size used, especially concerning the animal model, to the materials and methods section. In my opinion statistical analysis fo non parametric samples should be preferred in light of the low number of samples used for the statistical analysis. Please reply me discussing the reason for your statistical choice.
Author Response
Dear authors, I have a few comments on your paper. I think it is in general terms well written and presented.
We thank the Reviewer for the positive comment.
One of the major limitations of this work is the replicates number. I believe this limitation must be indicated in the discussion.
We agree with the Reviewer, however, from a minimum of 3 (very few), up to 6-7 replicates were used in each experiment, based on the similarity of the results. Indeed, we used a low number of mice when the differences between the genotypes were evident and supported by a small standard deviation. We actually did not feel to add this to the discussion, because it would have broken the flow; however, if it is necessary we will.
In addition, it would be appropriate to add the sample size used, especially concerning the animal model, to the materials and methods section.
We thank the Reviewer to raise this issue; we apologise for this mistake, we now added the sample size used in each Figure Legend, as indicated in the Material and Method session
In my opinion statistical analysis of non-parametric samples should be preferred in light of the low number of samples used for the statistical analysis. Please reply me discussing the reason for your statistical choice.
We agree with the Reviewer, however, given the low number of samples the non-parametric analysis might not be accurate, and we choose the parametric test because it is more powerful. We agree that increasing the sample size would be desirable, to establish the shape of data distribution, however, the differences observed were always very much appreciable, and the SD small, making us confident of the solidity of the data.